# Dietary Intake, Feeding Pattern, and Nutritional Status of Children with Cerebral Palsy in Rural Bangladesh

**DOI:** 10.3390/nu15194209

**Published:** 2023-09-29

**Authors:** Israt Jahan, Risad Sultana, Mousume Afroz, Mohammad Muhit, Nadia Badawi, Gulam Khandaker

**Affiliations:** 1CSF Global, Dhaka 1213, Bangladesh; sultana.risad@gmail.com (R.S.); mafroz.sumona@gmail.com (M.A.); mmuhit@hotmail.com (M.M.); gulam.khandaker@health.qld.gov.au (G.K.); 2Asian Institute of Disability and Development (AIDD), University of South Asia, Dhaka 1213, Bangladesh; 3School of Health, Medical and Applied Sciences, Central Queensland University, Rockhampton, QLD 4701, Australia; 4Cerebral Palsy Alliance Research Institute, Specialty of Child and Adolescent Health, Sydney Medical School, Faculty of Medicine and Health, The University of Sydney, Sydney, NSW 2006, Australia; nadia.badawi@health.nsw.gov.au; 5Grace Centre for Newborn Care, Children’s Hospital at Westmead, Sydney, NSW 2145, Australia; 6Discipline of Child and Adolescent Health, Sydney Medical School, The University of Sydney, Sydney, NSW 2006, Australia; 7Central Queensland Public Health Unit, Central Queensland Hospital and Health Service, Rockhampton, QLD 4700, Australia

**Keywords:** Cerebral Palsy, disability, diet, feeding problem, nutrition, underweight, child

## Abstract

(1) Background: Data on immediate causes of malnutrition among children with Cerebral Palsy (CP) are limited in low- and middle-income countries (LMICs). We aimed to assess the dietary intake pattern, feeding characteristics, and nutritional status of children with CP in Bangladesh; (2) Methods: We conducted a descriptive observational study in Shahjadpur, Bangladesh. Children with CP registered into the Bangladesh CP Register were included. Socio-demographic, clinical, dietary intake, feeding, gastro-intestinal conditions, and anthropometric data were collected. Descriptive and inferential statistics were reported; (3) Results: 75 children (mean (SD) age 3.6 (2.7) years, 42.7% female) and their caregivers participated. Overall, 53.6% and 46.4% of children were underweight and stunted, respectively. Two-thirds children consumed ≤4 out of 8 food groups. Meat, poultry, and fish; dairy products; and sugar consumption was lower among underweight children (43.4%, 48.8%, 25.0%) than others (56.7%, 51.2% 75.0%). Inappropriate feeding position was observed in 39.2% of children. Meal duration was >30 min/meal for 21.7‒28.0% children. Among all, 12.0% had feeding difficulties, 88.0% had ≥1 gastro-intestinal conditions; (4) Conclusions: The study reports preliminary data on the feeding characteristics, dietary intake, and nutritional status of children with CP in rural Bangladesh. The findings are crucial for cost-effective interventions, prevention, and management of malnutrition among children with CP in Bangladesh and other LMICs.

## 1. Introduction

Childhood disability and malnutrition are two global public health challenges which intersect and possess a perpetual relationship [1]. In the absence of early intervention, both disability and malnutrition can affect health, well-being, quality of life, and survival of children. The situation is more complex in low- and middle-income countries (LMICs) where most children with disability and malnutrition reside [2,3].

Cerebral Palsy (CP—one of the leading causes of childhood disability globally) is an umbrella term for a group of early-onset, non-progressive but permanent neurodevelopmental disorders caused by insult to the developing brain [4]. The prevalence of CP is estimated to be two to three times higher in LMICs than high-income countries (HICs) [5,6,7]. CP not only affects the movement or posture of a child, but the functional limitations also increase the risk of growth faltering [8,9]. In a recent systematic review, it was estimated that 40% of children with CP globally are malnourished [9]. However, the burden was estimated to be much higher in Bangladesh (two in three children with CP had at least one form of undernutrition) [10]. Though the high burden of malnutrition in children with CP has been documented in several studies, further evidence is required to understand the causal pathway and the perpetual relationship between these two major public health issues. It is well-established that undernutrition during early childhood (i.e., the first 1000 days of life) has a long lasting impact on the motor and cognitive development of children and these negative consequences could be reduced or averted with timely interventions [11,12,13]. At the same time, CP directly affects the functional status of children and their nutritional status by interfering in their dietary intake/feeding pattern, digestive system and metabolism [14,15]. 

Malnutrition is often the consequence of either illness or environmental/behavioral factors which affects the nutrient intake and balance in children [16]. The scenario is more complex for children with CP. Data on both direct (e.g., dietary intake, illness) and indirect (e.g., feeding characteristics, knowledge, socio-demographic and functional characteristics) contributors are crucial for a better understanding of the causal pathway [14]. Although the limited available evidence from Bangladesh and other LMICs identified low maternal education, severe gross motor function impairment, and presence of dysphagia as significant predictors of malnutrition in children with CP, the influence of these factors on the dietary intake, illness, and growth of children is rarely explored [14,17,18,19,20]. 

There is limited evidence available on nutrition intervention for children with CP in LMICs [21]. Lack of data as discussed in earlier section minimizes the scope for developing need-based intervention for prevention and management of malnutrition among children with CP in LMICs such as Bangladesh. In this study, we aimed to generate data on these crucial factors from a cohort of children with CP attending a rehabilitation center in rural Bangladesh.

## 2. Materials and Methods

A descriptive observational study was conducted between 1 January and 30 June 2023. Children with CP registered into the Bangladesh CP Register (BCPR) and their primary caregivers seeking consultation at the nutrition rehabilitation clinic of SMART CP center, Shahjadpur, were invited to participate. SMART CP centers are run by CSF Global and were established as part of a randomized controlled trial (RCT) in four sub-districts of Bangladesh (including Shahjadpur). The RCT aimed to improve the accessibility and utilization of early intervention and rehabilitation services for children with CP in rural Bangladesh.

### 2.1. Study Location and Settings

Shahjadpur is a northern sub-district in Bangladesh. It is located in Sirajganj district and is comprised of ~123,576 households with a total population of ~561,076 and ~232,037 child population aged <18 years [22]. As part of the ongoing services of Shahjadpur SMART CP center, we started a walk-in clinic for nutrition consultation and rehabilitation of children seeking services at the center. The walk-in clinic is operated once a week and is led by a team of a dietitian, a physiotherapist, and a community rehabilitation worker. 

Data were collected using a semi-structured questionnaire developed based on existing standard tools and pre-tested in the study area prior to finalization. The following information was collected.

#### 2.1.1. Dietary Intake Pattern

A 24-h recall method was used to collect information about the dietary intake pattern of the participating children. A dietitian interviewed all mothers/primary caregivers accompanying the child at the clinic to document detailed information about foods and beverages consumed by the child 24 h prior to the interviews.

#### 2.1.2. Feeding Characteristics

A semi-structured questionnaire was developed based on literature review and standard tools [23,24]. Information about food preparation, type and consistency of foods that the participating child consumed, feeding technique (e.g., position, utensils) used, time required to feed each meal, challenges faced during feeding, and signs of swallowing difficulties were documented. 

#### 2.1.3. Nutritional Status

Anthropometric measurements were collected by a community rehabilitation worker trained by the dietitian. Weight was measured following standard protocol adapted from World Health Organization (WHO) [25]. Height/length was measured for children who did not have any deformities/contracture and could stand independently. Knee height was measured for children who could not stand independently due to scoliosis. Mid-upper arm circumference (MUAC), bicep skinfold (BSF), triceps skinfold (TSF), and subscapular skinfold (SSF) thickness were measured for children aged ≤5 years. The collected measurements were compared with the WHO reference population to calculate z scores and classify the nutritional status of participating children. Detailed methods are available in our previous publications [10]. The caregiver’s perspective on the nutritional status of their children was also documented. 

#### 2.1.4. Gastro-Intestinal Complications and Feeding Difficulties

During the interview with the primary caregivers, the dietitian also collected information about signs of any feeding difficulties and history of any known gastro-intestinal complications among participating children. Medical records were reviewed if available.

#### 2.1.5. Predominant Motor Type, Topography, and Functional Classifications

A physiotherapist based at the SMART CP center assessed the predominant motor type, topography, and motor function of participating children. The predominant motor type and topography were documented following the BCPR protocol which uses the ACPR classification [7,26]. Gross motor function classification system (GMFCS) level was documented to assess the severity of motor function impairment [27]. The GMFCS classifies the gross motor skill of a child into five categories (GMFCS levels I, II, III, IV, and V) [27]. GMFCS I–II indicates a child can walk/sit without support from an assistive device whereas GMFCS III–V indicates that the child needs assistive devices to move around [27]. Manual Ability Classification Level (MACS) and mini-MACS were used to document the hand function of the participating children [28]. The MACS level is also categorized into five groups (MACS levels I, II, III, IV, V). 

#### 2.1.6. Data Extraction from BCPR

As mentioned earlier, all children recruited in this study were registered into the BCPR. In addition to the primary data collected in this study, we extracted data on selected variables from the BCPR. These are—(i) socio-demographic characteristics: information about housing characteristics, source of drinking water, type of toilet use, parental educational level, monthly family income. (ii) Associated impairments: data on presence and severity of associated impairments (e.g., vision, hearing, speech, intellectual, and epilepsy) were also extracted from the BCPR database [7]. The detailed data collection methods for these variables are available in our previous publications [7,26]. 

### 2.2. Data Management and Analysis

Z scores were calculated for the anthropometric measurements using WHO Anthro and WHO AnthroPlus software (v1.0.4). The z score for height-for-age (HAZ) was calculated for all children, the z score for weight-for-age was calculated for children aged ≤121 months, whereas the z scores for weight-for-height (WHZ), MUAC-for-age (MUACZ), BSF-for-age (BSFZ), TSF-for-age (TSFZ), and SSF-for-age (SSFZ) were calculated for children aged ≤61 months. Z scores between −2 SD and +2 SD were considered normal, a z score < −2 SD and >−3 SD was considered moderate undernutrition, z scores ≤ −3 SD were considered severe undernutrition, and a z score > +2 SD was considered overnutrition for all indicators mentioned above [25]. Food items consumed in the 24 h preceding the survey were grouped into eight major food groups following standard guideline [29]. The total number of food groups consumed (at least one serving was considered as minimum to be included in the calculation) was used as an indicator for food diversity. Descriptive (mean, SD, frequency, and proportion) and inferential statistics (chi-squared test, Fisher’s exact test, independent *t*-test) were used. Data management and all analyses were completed in SPSS version 25 (IBM Corporation, Chicago, IL, USA).

### 2.3. Ethics Approval

Ethics approval for this study was obtained from the Asian Institute of Disability and Development (AIDD) (ref no. Southasia-hrec-2022-12-02) and Bangladesh Medical Research Council (BMRC) (ref no. BMRC/NREC/2019-2022/36) in Dhaka, Bangladesh. 

## 3. Results

Between 1 January and 30 June 2023, 75 children received services from the walk-in clinic and were recruited into the study (mean (SD) age at recruitment: 3.6 (2.7) years, 42.7% are female). 

### 3.1. Socio-Demographic Characteristics

The socio-demographic characteristics of participating children are available in Appendix A. Overall, 73.0% of the children were aged ≤5 years. Most households had access to a sanitary latrine (65.2%) and all had access to a safe drinking water source. Overall, 97.1% mothers received at least some formal schooling, but only 7.2% were involved in any income generating activities (IGA). The mean (SD) age of the mothers and fathers of children (at the time of assessment/recruitment in the study) was 30.5 (7.4) and 36.6 (8.8), respectively. Fathers were the main earner in most families (98.6% were involved in IGAs). The median [IQR] monthly family income was 10,000 BDT [10,000, 15,000], approximately 100 USD [100, 150].

### 3.2. Clinical Characteristics

Among all children, 30.7% had a history of epilepsy, 34.7% had intellectual impairment, 10.7% had visual impairment, 9.3% had hearing impairment, and 62.7% had speech impairment. Most children (82.7%) had spastic CP, of which 87.1% had bilateral CP. Overall, 5.3% had dyskinesia and 4.0% had ataxia. GMFCS level III–V and MACS level III–V were documented in 84.7% and 74.3% children, respectively (Appendix A).

### 3.3. Nutritional Status of Participating Children

The median [IQR] of WAZ and HAZ was −2.1 [−3.1, −1.2] and −1.9 [−2.7, −0.7], respectively. Overall, 53.6% (*n* = 30/56) children were underweight, and 46.4% (*n* = 26/56) were stunted. Among children aged <5 years, the median [IQR] of WAZ, HAZ, WHZ, MUACZ, TSFZ, and SSFZ were −1.9 [−3.0, −1.2], −1.7 [−2.5, −0.7], −1.8 [−2.3, −1.0], −0.9 [−1.2, −0.4], −1.9 [−2.3, −1.2], and −1.9 [−2.3, −1.2], respectively. In this group, 46.3%, 42.5%, and 31.0% had underweight, stunting, and wasting according to WAZ, HAZ, and WHZ indicators, respectively. Furthermore, one child had severe acute malnutrition and *n* = 4 had moderate acute malnutrition according to the MUACZ indicator. Table 1 summarizes the nutritional status of participating children according to their anthropometric measurements.

The mean (SD) age at assessment was higher among underweight children compared to those with normal weight-for-age (4.1 (2.8) years vs. 2.6 (1.8) years respectively; *p* = 0.02). The mean (SD) age of the mothers was also significantly different between these two groups (33.1 (8.4) vs. 27.2 (6.1) years for underweight children vs. those who had a normal weight-for-age; *p* = 0.01). However, no significant difference between these two groups was observed in terms of housing type, source of drinking water, access to sanitation, maternal education, employment status and monthly family income, number of siblings, total household member, motor and clinical characteristics (e.g., predominant type, topography, presence of associated impairments, dysphagia, and reflux) (*p* > 0.05 for all) (Appendix A).

Overall, 73.3% (*n* = 55) caregivers thought their children were mildly underweight and 10.7% (*n* = 8) perceived that their children were severely underweight. When their responses were compared with the corresponding WAZ (*n* = 57), only 24.1% (*n* = 13/54, missing data = 3) caregivers had the correct perception about their child’s nutritional status. Irrespective of their perceived nutritional status, 71.6% (*n* = 53) caregivers thought that their child is not gaining weight adequately, and 93.2% (*n* = 69/74) mentioned that they are worried about the nutritional status of their children. (Table 2)

### 3.4. Feeding Characteristics

Most caregivers (77.3%) mentioned that their children cannot eat independently and need assistance with feeding. An additional 18.7% mentioned that their children can partially manage eating by themselves but sometimes require assistance. Furthermore, 92.0% of the mothers stated that they feed their children homemade foods, whereas an additional 6.7% feed their children both homemade and packaged/tinned foods. Most children ate medium thick/semi-liquid (30.7%) or thick (61.3%) foods. Consumption of liquid and semi-liquid food was comparatively higher among children with GMFCS level III–V (9.8% and 31.1%, respectively) compared to GMFCS level I–II (0.0% and 27.3%, respectively). 

Almost half of the caregivers did not feed their children in reclined position with adequate postural support. On average, 21.7–28.0% of mothers mentioned that they need >30 min to feed one meal to their children. A significant difference in mealtime was also observed between underweight children and children with normal nutritional status. The feeding characteristics of participating children are available in Table 3.

### 3.5. Food Consumption Pattern

The mean (SD) number of food groups consumed by participating children was 3.7 (1.2) out of 8 major food groups. More than two-thirds of the children (68.5%, *n* = 50/73) consumed ≤4 food groups. Among all, the majority consumed cereal, roots, and tubers (94.5%) and dairy products (71.2%). More than half of the participants consumed meat/poultry, fish, and eggs (fish and eggs were commonly consumed (66.7%, *n* = 28/42 and 40.5%, *n* = 17/42, respectively); however, none of them consumed organ meat. Only 30.1% children consumed vegetables at least once a day (of them 31.8%, *n* = *7*/22 children had consumed leafy vegetables) and 39.7% consumed fruits at least once in the preceding 24 h of the interview. Among those who ate fruits, the majority (86.2%, *n* = 25/29) consumed non-vitamin-A rich fruits (e.g., banana), 5.5% children consumed sugar as part of their diet, and 67.1% caregivers said they added oil to their children’s food. (Table 4) 

The number of food groups consumed was comparatively low among underweight children. Similarly, consumption of flesh foods, organ meat, eggs; dairy products; and sugar was slightly lower among underweight children compared to others (43.3% vs. 56.7%, 48.8% vs. 51.2%, and 25.0% vs. 75.0%, respectively); however, none of these differences were statistically significant (*p* > 0.05 for all) (Table 4).

### 3.6. Feeding Difficulties and Gastro-Intestinal Problems

Swallowing difficulties were reported in 12.0% of children (Appendix A). Over half (58.7%) of the caregivers mentioned that their children have difficulties in food bolus formation. Overall, 88.0% caregivers reported that their children had at least one gastro-intestinal/digestive issue in the two weeks preceding the interview. Among participating children, 65.3% had constipation, 49.3% had irregular bowel movements, 46.7% reported flatulence, and 14.7% reported indigestion. Furthermore, vegetable consumption was comparatively lower among children who had at least one gastro-intestinal/digestive issue than those without any such symptoms (26.7% vs. 55.6%, *p* = 0.08).

## 4. Discussion

The study reports preliminary data on feeding characteristics, dietary intake pattern, and nutritional status of children with CP in rural Bangladesh. Data reported here solely represent children and their caregivers who sought consultation at the rehabilitation center; thus, the findings may slightly vary from community settings and those who did not access any early intervention or rehabilitation services in the study area. Our data suggest that almost half of the participating children were undernourished. Low diversity in food consumption, long mealtimes, and inappropriate feeding techniques were commonly observed in most of the undernourished children, compromising their dietary intake and nutritional status. Gastro-intestinal conditions were also commonly reported. The findings provide initial insight to these interrelated issues that directly affect the nutritional status of children with CP. The study therefore provides important data to design nutrition intervention for the prevention and management of malnutrition among children with CP in rural areas of Bangladesh and similar economic/cultural settings. 

The sociodemographic characteristics of our study participants were slightly higher than those observed in the BCPR cohort [7]. Maternal literacy and monthly family income were also comparatively higher among the participating children when compared to other families registered into the BCPR [7]. Both these factors are known predictors of malnutrition among children with CP in Bangladesh [10,17]. Almost half of the children in our study were underweight and/or stunted. This percentage is substantially higher than reported in general population. The latest Bangladesh Demographic and Health Survey (2022) showed that about 22%, 24%, and 11% children aged below 5 years in Bangladesh have underweight, stunting, and wasting, respectively [30], whereas in our cohort (i.e., children with CP aged <5 years), these percentages were found to be 46.3%, 42.5%, and 31.0%, respectively. Nevertheless, the nutritional status of this cohort is slightly better than observed in the wider group of BCPR or studies from other LMICs [17]. Socioeconomic status directly influences an individual’s or household’s food consumption pattern as well as the nutritional status of an individual. The slightly better sociodemographic characteristics and monthly family income observed in our study cohort compared to the wider group of BCPR is likely to have a positive impact on the nutritional status of participating children [17]. Furthermore, most children in our cohort were aged five years or less. Data from other studies in LMICs show that in absence of early intervention and rehabilitation, growth faltering becomes more common among children with CP, mostly due to the increased level of motor impairment as they grow up [31,32]. It is therefore crucial to provide need-based intervention at an early stage to prevent malnutrition among children with CP in LMICs such as Bangladesh. 

Inadequate feeding practices were commonly observed among children in our study. Our data suggest that most children in our cohort had a cereal-based diet with less frequent consumption of plant- or animal-sourced protein, vegetables, and fruits. Though in our study we could not perform nutritional analysis of their diet, the food consumption pattern observed indicates that many of our participating children may not have met their recommended dietary intake. Studies conducted in other countries also reported similar observations [33,34]. In the absence of a balanced diet, these children are at high risk of growth faltering, malnutrition, and compromised immunity. Furthermore, constipation and gastro-intestinal conditions are commonly observed among children with CP due to limited gastrointestinal motility and alteration in the digestive system associated with functional limitations. Adding vegetables and fruits is essential for gut health to relieve some of the symptoms and improve bowel movement among these children [35]. The high constipation rate, especially among study participants who did not have vegetables or fruits in their diet, also supports the above finding. Similar findings were observed in other studies [20,36]. These practices could be improved through education and demonstration sessions and empowering the caregivers to select/plan a balanced diet for their children when preparing meals or feeding them. 

About one-third of our participating children were fed in an inappropriate position (e.g., lying/prone position, head tilted to the back or side of the body without postural support) which is concerning as it increases the risk of aspiration pneumonia. Pneumonia was identified as one of the leading causes of premature death among children with CP in Bangladesh [37]. Furthermore, a large number of children in our study also consumed liquid and semi-liquid food irrespective of the presence/absence of dysphagia. This may reduce the calorie and nutrient density of the consumed food, thus limiting the calorie and nutrient intake of participating children. The long meal duration as observed in our study is also concerning as long mealtime has been previously identified as a predictor for low quality-of-life and inadequate dietary intake pattern among children with CP in other LMICs [38,39,40]. We also observed a significantly longer feeding duration during the two main meals, i.e., breakfast and dinner time among underweight children than those who had a normal weight-for-age. Long meal duration could be an indication of oral and pharyngeal conditions in children which affect their swallowing capacity and the need for modification of the feeding methods (e.g., position, texture and consistency of the food, responsive feeding). There is a need for detailed investigation to identify the key reasons and take necessary actions to ensure optimal dietary intake. Nonetheless, as mentioned previously, long meal duration is stressful for both caregivers and children and it has a negative impact on the quality-of-life of caregivers and dietary intake of children with CP [24]. 

Childhood malnutrition is a cause and consequence of humanitarian crises in many parts of the world. However, for children with CP in LMICs, malnutrition remained a neglected crisis without any attention and lack of international initiatives. A comprehensive program to ensure and promote healthy eating and address the underlying risk factors of malnutrition is essential to improve the nutritional outcome of children with CP in LMICs. The involvement of a multidisciplinary team (including dietitian, occupational therapist, speech therapist, and medical professional) is required when developing a nutritional management plan for this vulnerable group of population [41]. Intervention should emphasize: (i) improving dietary practices, (ii) improving feeding behavior and caring practices, and (iii) ensuring treatment of the underlying conditions that increase their risk of malnutrition (e.g., infection, gastro-intestinal disorder) [42]. However, in LMICs such as Bangladesh, the challenge remains the provision of such multidimensional services, especially in rural areas/community-based settings, and the allocation of logistics/resources to ensure the receipt of need-based and targeted intervention. Community-based programs to enable mothers/caregivers and healthcare professionals in the nutritional management of children with CP could be useful. In a previous study in Bangladesh, teaching mothers or caregivers of children with CP on feeding practices improved the dietary intake pattern and feeding position/characteristics of participating children [23]. Similarly, community-based studies in Tanzania and Ghana found positive impacts of caregiver training and provision of supporting equipment/assistive devices on feeding characteristics quality of diet and nutritional status of children with CP, and the quality of life of their mothers [40,42]. 

Though the study has provided insights into some of the very important factors that affect the nutritional status of children with CP, our study has several limitations. First, the smaller sample size and center-based participant recruitment compromise the generalizability of the findings. Second, we used the 24 h recall method to collect information about dietary intake patterns. Although the 24 h recall method is a validated tool to assess dietary intake pattern, it may not represent the habitual diet in children. Furthermore, due to inadequate data, we could not estimate the calorie and nutrient intake of the participants and had to only report the diversity of consumed food items. Finally, due to the smaller sample size, we only provided descriptive statistics and could not establish causal relationships between these factors. 

## 5. Conclusions

It is evident that children with CP are susceptible to malnutrition. Studies conducted in LMICs have established the relationship between motor function limitations and malnutrition among children with CP in low-resource settings. However, data on direct factors such as dietary intake and feeding patterns are limited. Our study findings therefore add great value to the existing literature to understand the underlying factors of malnutrition among children with CP in LMICs such as Bangladesh. The findings have already identified a few gaps where targeted intervention could improve the nutritional outcome of the participating children. These findings are highly relevant to strategic planning and need-based intervention for the prevention and management of malnutrition among children with CP in Bangladesh and other similar low-resource settings.

## Figures and Tables

**Table 1 nutrients-15-04209-t001:** Nutritional status of participating children.

Characteristics	WAZ	HAZ	WHZ	MUACZ	TSFZ	SSFZ	BAZ
*n*	57	57	51	41	41	41	45
Median	−2.1 [−3.1, −1.2]	−1.9 [−2.7, −0.7]	−1.8 [−2.5, −1.0]	−0.9 [−1.2, −0.4]	−1.9 [−2.3, −1.2]	−1.9 [−2.3, −1.2]	−1.5 [−2.0, −0.8]
Nutritional status, *n* (%)							
Overnutrition ^1^	1 (1.8)	1 (1.8)	0 (0.0)	0 (0.0)	0 (0.0)	0 (0.0)	0 (0.0)
Normal ^2^	26 (45.6)	30 (52.6)	30 (58.8)	36 (87.8)	23 (56.1)	21 (51.2)	34 (75.6)
Moderate undernutrition ^3^	13 (22.8)	13 (22.8)	14 (27.5)	4 (9.8)	15 (36.6)	15 (36.6)	8 (17.8)
Severe undernutrition ^4^	17 (29.8)	13 (22.8)	7 (13.7)	1 (2.4)	3 (7.3)	5 (12.2)	3 (6.7)

^1^ z score > 2.0, ^2^ z score −2.0 to +2.0, ^3^ z score −2.1 to <−3.0, ^4^ z score ≥ 3.0.

**Table 2 nutrients-15-04209-t002:** Caregiver’s perspective about their child’s nutritional status.

Characteristics	*n* (%)
Nutritional status, *n* = 75	
Overweight	1 (1.3)
Normal weight	11 (14.7)
Mildly underweight	55 (73.3)
Severely underweight	8 (10.7)
Child is gaining weight adequately, *n* = 74	
No	53 (71.6)
Yes	8 (10.8)
Don’t Know	13 (17.6)
Is worried about child’s nutritional status, *n* = 74	
No	3 (4.0)
Yes	69 (93.2)
Do Not Know	2 (2.7)

**Table 3 nutrients-15-04209-t003:** Feeding characteristics of participating children.

Feeding Characteristics	*n* (%)	Nutritional Status, *n* = 56, *n* (%)		*p* Value ^2^
		Underweight, *n* = 30	Normal, *n* = 26	
Feeding position, *n* = 74 ^1^				
Inappropriate ^3^	29 (39.2)	12 (40.0)	6 (23.1)	0.18 ^4^
Appropriate ^3^	45 (60.8)	18 (60.0)	20 (76.9)	
Feeding type, *n* = 75				
Self-fed	3 (4.0)	1 (3.3)	1 (3.8)	0.99
Need to be fed by others	58 (77.3)	23 (76.7)	20 (76.9)	
Can partially manage eating by self	14 (18.7)	6 (20.0)	5 (19.2)	
Tube-fed	0 (0.0)	0 (0.0)	0 (0.0)	
Food preparation, *n* = 75				
Homemade	69 (92.0)	27 (90.0)	24 (92.3)	0.39
Processed/packaged	1 (1.3)	0 (0.0)	1 (3.8)	
Both homemade and processed	5 (6.7)	3 (10.0)	1 (3.8)	
Food consistency, *n* = 75				
Liquid	6 (8.0)	3 (10.0)	3 (11.5)	0.40
Semi-liquid/medium thick	23 (30.7)	12 (40.0)	6 (23.1)	
Thick	46 (61.3)	15 (50.0)	17 (65.4)	
Mealtime				
Morning, *n* = 75				
≤30 min	54 (72.0)	18 (60.0)	25 (96.2)	0.001
>30 min	21 (28.0)	12 (40.0)	1 (3.8)	
Mid-morning snacks, *n* = 75				
≤30 min	57 (76.0)	21 (70.0)	25 (96.2)	0.01
>30 min	18 (24.0)	9 (30.0)	1 (3.8)	
Lunch, *n* = 73				
≤30 min	47 (62.7)	19 (63.3)	21 (80.8)	0.23
>30 min	13 (17.3)	5 (16.7)	1 (3.8)	
N/A ^5^	15 (20.0)	6 (20.0)	4 (15.4)	
Evening snacks, *n* = 75				
≤30 min	23 (30.7)	11 (36.7)	7 (26.9)	0.42
>30 min	8 (10.7)	3 (10.0)	1 (3.8)	
N/A ^5^	44 (58.6)	16 (53.3)	18 (69.2)	
Dinner, *n* = 74				
≤30 min	56 (74.7)	21 (70.0)	24 (92.3)	0.02
>30 min	18 (24.0)	9 (30.0)	1 (3.8)	
N/A ^5^	1 (1.3)	0 (0.0)	1 (3.8)	

^1^ Missing data; ^2^ Fisher’s exact test; ^3^ Feeding position was categorized inappropriate if the child was fed on lying/prone position and/or with head tilted to the back or side, whereas feeding position was categorized as appropriate if a child was fed on reclined position with adequate back support; ^4^ Chi-squared test; ^5^ N/A indicates those who skipped this meal.

**Table 4 nutrients-15-04209-t004:** Food consumption pattern.

Major Food Groups	Total ^1^	Normal WAZ (*n* = 25) ^2,3,4^	Underweight (*n* = 29) ^2,3,4^	*p* Value ^5^
Cereal, roots and tubers	69 (94.5)	23 (46.0)	27 (54.0)	0.88
Legumes and nuts	8 (11.0)	3 (60.0)	2 (40.0)	0.52 ^6^
Milk and milk products	52 (71.2)	21 (51.2)	20 (48.8)	0.20
Flesh foods, organ meat, eggs	42 (57.5)	17 (56.7)	13 (43.3)	0.09
Fish and fish products	28	11	8	Na
Meat and poultry	7	2	2	Na
Organ meat	0	0	0	Na
Eggs	17	10	5	Na
Vegetables	22 (30.1)	2 (18.2)	9 (81.8)	0.04 ^6^
Green leafy vegetables	7	2	2	Na
Vitamin A rich tubers	0	0	0	Na
Other vegetables	17	2	7	Na
Fruits	29 (39.7)	9 (47.4)	10 (52.6)	0.91
Vitamin A rich fruits	5	1	1	Na
Other fruits	25	8	10	Na
Sweet	4 (5.5)	3 (75.0)	1 (25.0)	0.25 ^6^
Oils	49 (67.1)	18 (50.0)	18 (50.0)	0.44

^1^ Represents those who consumed from specific food groups; ^2^ Missing data; ^3^ Row percentage; ^4^ WAZ (i.e., underweight/normal weight) was calculated for children aged ≤121 months; ^5^ Chi-squared test; ^6^ Fishers exact test.

## Data Availability

The data presented in this study are available on request from the corresponding author. The data are not publicly available due to privacy/ethical restrictions.

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
