# Peer review of "Dietary Intake, Feeding Pattern, and Nutritional Status of Children with Cerebral Palsy in Rural Bangladesh"

_nutrients, 2023, doi:10.3390/nu15194209_

Round 1

Reviewer 1 Report

In this manuscript, the authors utilized the 24-hour recall method to analyze data on dietary intake, feeding patterns, and the nutritional status of children with cerebral palsy. The aim was to understand the relationship between childhood disability and malnutrition in low- and middle-income countries.

Overall, the manuscript is well-written and effectively summarizes their findings. The authors report that almost half of the subjects were undernourished, which was associated with low diversity in food consumption and long mealtimes. I have some minor comments.

Page 5, lines 202-204

It might be interesting to discuss whether the ages of mothers and caregivers, as well as the number of children with or without CP they have cared for before, could have affected the results.

Page 6, lines 223-224

Did the difference in mealtime result from variations in the total amount (grams or calories) of meals and food groups served by mothers and caregivers? Table 3 shows that while the majority of children were not applicable and the difference was not statistically significant, there was a trend indicating that more underweight children consumed evening snacks in less than 30 minutes compared to normal children. The authors could discuss this discrepancy.

Other minor comments:

Page 2, lines 87-88

It's acceptable to provide a detailed explanation of the sentence (and sustain the activities...), but it represents unnecessary information for this study. Similarly, mentioning adults may be confusing to readers since they are not the subject of this study (Page 3, line 94).

Page 6, lines 217-220

It would be helpful for readers to provide a brief description of the major differences between GMFCS levels I-II and III-V.

Reviewer 2 Report

I find it a highly interesting work and I encourage you to continue with this line to be able to improve the conditions of these children in their lives and in the orientation of their diet. 

Author Response

Thank you very much for taking the time to review this manuscript and for your positive comments. 

Reviewer 3 Report

Cerebral palsy in children and malnutrition has always been a problem. Particularly if very low-income countries are affected. This paper deals with children in Bangladesh dietary data, nutritional status, income, dietary options with nutritional ratios were analysed.   The results of the study come as no surprise to me. And I am not concerned here whether these are children with cerebral palsy or healthy children. These problems apply to other countries as well. the most important conclusions in this manuscript are how to improve humanitarian aid, how to improve parental education, how to improve the lives of these young patients. I suggest that the paper be accompanied by a few more sentences on how to make these changes in these countries. Just writing and saying nothing will not help. There has to be a scheme of action.  

The manuscript should also be based on how to properly nourish children and supplement with the right dose of vitamins and improve the gut microbiome. this will increase the improvement of the work and there should be a little more references
